# Reassessment of Evidence about Coinfection of Chickenpox and Monkeypox (Mpox) in African Children

**DOI:** 10.3390/v14122800

**Published:** 2022-12-15

**Authors:** Ola Khallafallah, Charles Grose

**Affiliations:** Division of Infectious Diseases, Virology Laboratory, Department of Pediatrics, University Hospital/Room BT2001, University of Iowa, 200 Hawkins Drive, Iowa City, IA 52242, USA

**Keywords:** varicella-zoster virus, herpesvirus, variola, mousepox, orthopox virus, murid herpesvirus-4, interleukin-6

## Abstract

In west and central Africa, monkeypox occurs mainly in older children, adolescents and young adults. In two large epidemiology studies of monkeypox outbreaks, the investigators observed a sizable number of coinfections of chickenpox (varicella) and monkeypox. Based on a review of the literature, we propose that chickenpox (human herpesvirus-3 infection) is a risk factor for acquisition of monkeypox infection. Our hypothesis states that the chickenpox skin lesion provides an entry site for the monkeypox virus, which is harbored on a fomite in the environment of the patient. The fact that monkeypox can enter via a scratch or abrasion is a known mechanism of spread for three other poxviruses, including mousepox (ectromelia), orf and molluscum contagiosum. There are many similarities in pathogenesis between certain poxviruses and chickenpox, including a viremia with a cellular stress response leading to high levels of the IL-6 cytokine. One very revealing observation in the two epidemiology studies was that the number of pox as well as the severity of disease in children with chickenpox and monkeypox coinfection was not greater than found in children with monkeypox alone. Based on the above observations, we conclude that, when chickenpox precedes monkeypox, priming of the immune system by the earlier chickenpox infection moderates the severity of the secondary infection with monkeypox. This conclusion also has important public health implications about chickenpox surveillance.

## 1. Introduction

Recent reports have described coinfection of chickenpox (varicella) with monkeypox (MPX) among children in African countries [1,2]. Investigators in the earlier reports have generally favored an explanation that monkeypox likely preceded chickenpox in the children. Based on our review of the literature as well as more recently published findings about varicella pathogenesis, we propose the opposite: chickenpox in children is a risk factor for subsequent acquisition of monkeypox. We cite several arguments in favor of this hypothesis. These include the following: (i) similarities with pathogenesis of mousepox (ectromelia), (ii) similarities with pathogenesis of Orf, (iii) similarities with pathogenesis of molluscum contagiosum, and (iv) research findings about similarities in the pathogenesis of chickenpox with certain poxviruses [3,4]. If varicella is a risk factor, then our hypothesis implies that eradication of monkeypox may be an even more difficult task in children living in central and west Africa. Thus, this review also has public health implications. Finally, we note that the World Health Organization recommended in late 2022 that the word monkeypox be replaced by mpox.

## 2. Etymology Surrounding Chickenpox and Varicella

The word varicella is actually an irregular diminutive form of the word variola [5]. By the 18th century in England, it was recognized that there was a less virulent vesiculopustular variant of what appeared to be variola and this more benign disease was called varicella. The word chickenpox was in use in England by the 17th century and again referred to an assumed milder form of smallpox; the name chicken is not an avian allusion but taken from a Middle French word ‘chiche or chicke’ describing a chickpea [5]. The virus causing varicella (chickenpox) was characterized as a herpesvirus known as varicella-zoster virus (VZV) or human herpesvirus-3 and not a poxvirus in the mid-20th century [6]. The Nobel laureate Weller first grew varicella in cultured cells [6]. It is striking therefore to observe in the 21st century that coinfection of varicella and another poxvirus very similar to variola, namely monkeypox, can occur in children and the exanthems are very overlapping in character [7].

## 3. Reports of Coinfection in Africa

### 3.1. Analysis of 151 Coinfections

A surveillance program conducted in the Sankuru district of the Democratic Republic of the Congo (DRC; formerly called Zaire) between 2006–2007 identified 151 cases of coinfection between monkeypox and chickenpox, among a total of 1158 suspected cases of monkeypox [1]. The case definition by the DRC Ministry of Health included any person with a sudden onset of high fever, followed a few days later by a vesiculopustular rash, presenting on the face, palms on the hands and soles of the feet. Vesicular fluid and/or scabs were collected from each patient and processed for analysis by PCR assays with specific primers for both monkeypox and chickenpox. To increase specificity even further, a positive varicella PCR test was confirmed by a second varicella PCR test with a different set of primers. 

The average age of children in the MPX alone group was around 10 years, the MPX/VZV cases around 11 years, while the VZV cases alone were older (around 17 years). In patients 30 years and older, almost all were VZV alone. The average number and range of skin lesions were the same in all three disease profiles. However, this survey made an interesting discovery that coinfected children had significantly lower likelihood of fever, swollen cervical lymph nodes and pox on the hands/soles than children with MPX alone. With regard to geography, MPX alone occurred significantly more often in small villages while VZV alone was more common in cities; MPX/VZV were intermediate. These data are similar to data collected in the United States a half-century ago [8]. The latter epidemiology data showed that chickenpox outbreaks in children occurred annually in cities but only every 2–3 years in small villages.

### 3.2. Analysis of 134 Coinfections

A surveillance program conducted in the Tshuapa province of the DRG identified 1271 suspected cases of monkeypox between September 2009 and December 2014 [2]. Both skin lesion swabs and crusts were collected from each enrolled patient. All samples were first tested in the DRG by an MPX-specific PCR test; all samples including the negative samples were sent to the Centers for Disease Control in the United States, where a second PCR test for MPX was performed as well as a PCR test for VZV on all MPX-positive and MPX-negative samples. Among these samples, 400 were MPX only, 457 were VZV only and 134 were found to be coinfections with both monkeypox and chickenpox. 

In this study, the average age for both MPX alone and MPX/VZV was 15 years while that for VZV alone was 18. The pox lesions were quantified into 4 groupings: benign (1–25), moderate (26–100), grave (101–250) and grave plus (>250). The average number of skin lesions was similar between MPX alone (143) and MPX/VZV (130), while that for VZV alone was 72, significantly less than either of the other two groups. In general, the VZV/MPX group was less ill than the MPX group alone; for example, the measured signs and symptoms including cough, mouth sores, sore throat, lymphadenopathy, and bed-ridden status were significantly less in the VZV/MPX group. In turn, both MPX alone and MPX/VZV had more signs and symptoms than VZV alone. 

## 4. Vignettes about the Pathogenesis of Three Poxviruses

Our hypothesis is based on the epidemiology and pathogenesis of three different poxviruses, as well as the pathogenesis of a bacterial super-infection in children with varicella. 

### 4.1. Mousepox

Mousepox is also known as ectromelia. The genus *Orthopox* virus includes vaccinia virus, cowpox virus, monkeypox and mousepox (ectromelia). Ectromelia was first identified in a laboratory mouse in 1930. The well-known Australian virologist Fenner used ectromelia infection to form his frequently cited model of viral pathogenesis [9,10]. One of the Authors of this Review wrote a Commentary about mousepox and chickenpox in 1981, based on the Fenner model of a dual viremia pathogenesis (Figure 1) [3]. The average incubation period for varicella is considered to be 14 days, based on numerous observational studies. 

Mousepox is easily spread from animal to animal in a small animal care facility (vivarium) if just one animal is infected. The virus in the skin lesions is transferred to fomites, such as the bedding in the cages which house multiple mice. Thereafter the uninfected mice quickly contract the viral infection via small breaks in the skin. A dual viremia ensues, followed by an exanthem. Ectromelia does not spread to humans. 

### 4.2. Orf

The Orf virus is a member of the *Parapoxvirus* genus [11,12]. The name Orf is derived from an Old English word that means ‘livestock.’ The virus infection primarily occurs in sheep and goats, where it causes small pustules in the skin. The infection has been called thistle disease because the prickles (spiny protrusions) on a thistle plant can scrape off bits of the pustules from an infected animal as that animal grazes in a pasture. When uninfected sheep pass through the same patch of thistles, they in turn are scratched by the prickles and then inoculated with the virus into their epidermis. The thistle plant acts as a fomite. Alternatively, farm workers who care for the animals can contract Orf, especially on their hands. The virus enters the hands of the human handlers through small breaks in the skin, as can be seen in chickenpox lesions The skin lesion has the appearance of a pustule on the hand. Under these circumstances, Orf is a zoonosis, sometimes called contagious ecthyma.

### 4.3. Molluscum Contagiosum

This poxvirus is a member of the genus *Molluscipoxvirus* [13,14]. The skin disease usually occurs in younger children and is benign. Transmission of the Mollusca can occur by direct skin-to-skin contact, for example, between children who are wrestling. Transmission is more commonly associated with events such a bathing within a home in a tub shared by siblings or swimming in a community swimming pool. In these settings, transmission is often associated with shared usage of a towel or sponge among several children, one of whom has molluscum contagiosum. The virus enters from the fomite through small abrasions or breaks in the skin, often on the feet. There is no known animal host. 

Thus, the epidemiology and pathogenesis of the above poxviruses from three different genera clearly show that all three can be easily transmitted from a fomite to an uninfected host, via small breaks in the skin. The viruses are stable for hours at ambient temperature.

## 5. Vignettes about the Pathogenesis of Chickenpox

### 5.1. Later Occurrence of Chickenpox in Tropical Countries

For reasons still not completely understood, chickenpox in tropical countries is not a disease of early childhood, as is the case in temperature climates [15,16,17]. In the United States, for example, chickenpox commonly occurred in outbreaks among young children as they entered kindergarten and the first grade [8,18]. The outbreaks occurred annually in schools in large cities and perhaps every 2–3 years in schools in smaller towns and rural villages. In the United States, with an annual birth cohort of around 3 million in the late 20th century (before universal varicella vaccination), the mortality rate was around 100–150 deaths per year in young children. On the other hand, in tropical countries without universal varicella vaccination, chickenpox typically occurs in late childhood, adolescence and early adulthood. Again, the mortality rate is lower than monkeypox.

### 5.2. Elevated IL-6 Transcription and Protein Production

There is no small animal model for VZV infection, because of the strong tropism of VZV for human tissues. To overcome that obstacle, a model was developed in which explants of human newborn tissues are grown in cell culture; the model is called “skin organ culture” (SOC) [19]. The explants, which remain viable for 3–4 weeks, accurately represent VZV infection of the skin in a child. To investigate innate immune transcription in response to VZV infection, a PCR array that measured 84 different markers was performed on VZV-infected SOC [20]. Of importance, IL-6 was the single most elevated transcript in the profile of VZV-infected human skin. To validate the above microarray result, an IL-6-specifc PCR assay was performed with a similar result. Finally, to confirm the transcription data, levels of the IL-6 protein were measured in the medium overlying an infected SOC. The IL-6 levels exceeded 30,000 pg/mL. Because the half-life of IL-6 is less than one day, the above high IL-6 level cannot represent accumulation of IL-6 in the medium. 

### 5.3. Sequential Herpesvirus and Pneumovirus Infection

There is an informative example where a prior herpesvirus infection improves the immune response to and lessens the severity of a second virus infection [21]. The animal model system includes infection of mice with mouse pneumovirus 28 days after mice were infected with murid herpesvirus-4 (herpesvirus 68). Murid herpesvirus-4 is similar to human Epstein–Barr herpesvirus and mouse pneumovirus is similar to human respiratory syncytial virus. The authors selected the word ‘priming’ to describe the effect of an earlier Murid herpesvirus-4 infection; in short, priming with a herpesvirus infection provided pronounced protection against subsequent severe pneumoviral disease in mice. Priming significantly expanded the pool of activated CD8+ lymphocytes. Priming also modified the IL-6 response to pneumoviral infection. The latter effect appeared more related to the timing of the IL-6 response in the doubly infected mouse rather than the absolute peak of IL-6 secretion.

## 6. Denouement

### 6.1. Comparison with Three Other Poxviral Diseases

The above three examples of other poxvirus infections clearly show that a poxvirus can be transferred from an infected animal via a fomite to an uninfected animal or indeed to an uninfected human, through breaks in the skin of the animal or human subject. If that subject were a human with chickenpox, each of the chickenpox vesicles would be a conduit whereby a poxvirus could easily bypass the normal skin barriers and enter the epidermis, to begin a replication cycle with a subsequent viremia (Figure 1). The fact that the average age of children who acquired monkeypox in the above two reports from the DRC was 10–15 years may suggest that the index case of monkeypox occurred in an adolescent with chickenpox; presumably the inoculum of monkeypox required for infection would be less if the virus entered via a chickenpox skin lesion. The reservoir of infection is considered to be a rodent, although the host species does not appear to be the same in different outbreaks [22,23]. 

### 6.2. The Role of IL-6 Expression in Virus-Infected Tissues

Based on our interpretation of the thorough data collected from the children in the DRC, the varicella infection occurred before onset of the monkeypox infection in the children with coinfection (Figure 1). What is striking is the similarity of the pathogenesis model for chickenpox and poxviruses such as monkeypox. Both include a viremia followed by an exanthem as virus exits the bloodstream and enters the epidermis. Based on detailed studies of the immune signature of patients infected with monkeypox, we know that high levels of IL-6 have been documented in the blood, for example, levels as high as 10,000 pg/mL [24,25].

IL-6 is known to be an important factor in the pathogenesis of monkeypox infection based on an animal model. In a study performed in long-tailed macaque monkeys inoculated with monkeypox virus, numerous samples were obtained on days 2, 4, 6, 8, 10 and 12 after infection [26]. The fever peaked on day 6 and pox lesions appeared on the skin following a viremia; in a sacrificed animal, virus was found in tonsils, lung, lymph nodes, spleen and colon. By day 10, an IgG antibody response was detectable and by day 12, there were signs of clinical recovery. With regard to measurement of cytokines in the blood, the peak level of IL-6 was obtained on day 8. The authors concluded that IL-6, together with Interferon-gamma and IL-1ra, played a role in the response of the macaques to monkeypox infection. 

## 7. Conclusions

Based on the publications cited above, we propose that a preponderance of evidence suggests that chickenpox is a risk factor for contracting monkeypox in children (Figure 1). Specifically, the varicella skin vesicle provides an entry site for monkeypox residing on a fomite in a child’s environment [27]. Based in part on the murine herpesvirus/pneumovirus model, an equally important second point is the hypothesis that the prior varicella infection in children reduces the severity of monkeypox because varicella infection primes the immune response. The above hypotheses answer the query from prior papers why children infected with both monkeypox and chickenpox did not have more pox and more severe disease generally than children infected with only monkeypox.

We question the speculation in earlier reports that monkeypox precipitated reactivation of a latent varicella infection from the dorsal root ganglia and therefore the coinfection represented herpes zoster and monkeypox. Reports from the older smallpox literature in the United States by established consultants in pediatric infectious diseases did not mention herpes zoster as a complication of smallpox in children [28]. The epidemiology of herpes zoster has been well studied in many countries around the world [29,30,31,32]. There are no examples of major outbreaks of herpes zoster among previously healthy children. Short courses of high-dosage corticosteroids are the most common risk factor for onset of herpes zoster in individual children with a prior history of chickenpox, especially when associated with the onset of chemotherapy for a newly diagnosed cancer or rheumatological disorder [30]. Herpes zoster also has been associated in rare instances with long-term environmental chemical exposure in abandoned refuse dumps; therefore, it is possible but seemingly unlikely that children in the DRC in two different geographic regions were exposed to a previously unrecognized toxin, such as organochlorine pesticides [33]. Nevertheless, even though there are limited data about herpes zoster in African countries, to have 134 cases or 151 cases of herpes zoster within a short time period in residents of a village in the DRC has no medical precedent [34].

We note that there has been considerable social and political disruption in DRC recent history; this turmoil in turn has led to low vaccination rates against several childhood diseases [35]. Outbreaks of measles, in particular, could lead to an enhanced susceptibility to infection overall in children [36]. However, recent measles outbreaks were not mentioned in either paper describing coinfection, and measles-induced immunosuppression would not lead preferentially to coinfection. A recent systematic review of the epidemiology of monkeypox analyzed the data from 71 documents [22]. They concluded that monkeypox was becoming a more common disease in the African countries of the DRC, the Republic of the Congo, the Central African Republic, Nigeria, Gabon, Cameroon, Liberia, Sierra Leone and the Cote d’Ivoire. In a short paragraph in the Discussion, they suggested that surveillance for varicella be strengthened during the investigations of monkeypox outbreaks, because of the differing public health implications regarding monkeypox alone, varicella alone or coinfections with varicella and monkeypox [22]. 

We cannot exclude the possibility that strains within the two clades of monkeypox may interact differently during a coinfection with varicella virus [37,38]. Nevertheless, we concur with an earlier suggestion that evidence for concurrent chickenpox be sought in major outbreaks of monkeypox among children and adolescents in Africa [2].

## Figures and Tables

**Figure 1 viruses-14-02800-f001:**
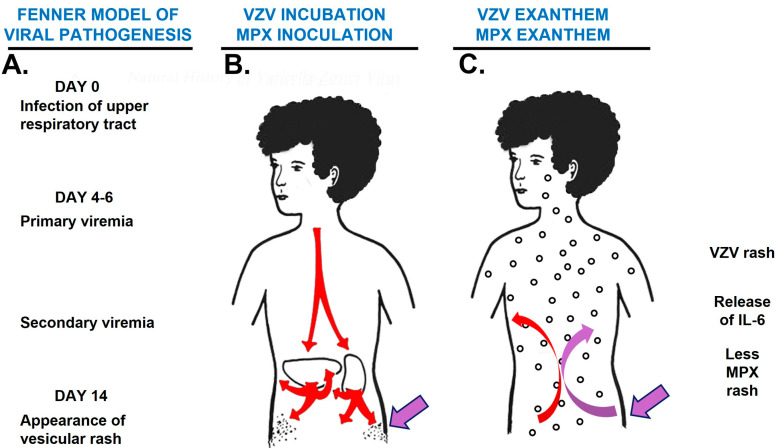
Pathogenesis of chickenpox (varicella) alone and coinfection of varicella and monkeypox. (**A**) Fenner model of varicella pathogenesis. The model includes two viremic episodes over a period of 14 days, after which the exanthem first appears. (**B**) A schematic drawing of the Fenner model. The varicella viremias are high-lighted with red arrows inside the body. During a viremia, the vesicles appear around 14 days after the first day of infection. The purple arrow on the left flank illustrates a potential site for monkeypox virus to enter through the epidermis within a varicella vesicle. (**C**) A schematic drawing of the dual exanthems following coinfection with varicella and monkeypox. The varicella viremia is drawn with red arrows and the monkeypox viremia is drawn with purple arrows inside the body. New pox lesions appear over a period of 5–7 days. The IL-6 marker indicates the release of IL-6 following varicella infection. The marker for less MPX rash means less rash that expected in a dual chickenpox and monkey infection. Concepts for this figure are derived from an earlier figure by an author of this publication [3]. Abbreviations: VZV, chickenpox (varicella); MPX, monkeypox.

## Data Availability

Not applicable.

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
