# Peer review of "Reassessment of Evidence about Coinfection of Chickenpox and Monkeypox (Mpox) in African Children"

_viruses, 2022, doi:10.3390/v14122800_

Round 1
Reviewer 1 Report
The manuscript by Kallafallah and Grose is quite informative and brings potentially new perspective to the primary acquisition of VZV followed by monkeypox, especially looking at the priming infection as both a source for acquiring the latter and a way to abate the effects of the latter.
The manuscript makes cogent arguments behind the authors’ hypotheses, and provides adequate circumstantial evidence for their ideas. There is nothing technical on which to comment because the ideas are entirely plausible. However, I am including a small list of notes for the authors and/or copy editors to address, small problems I caught that are easily remedied. Once these are fixed, there is no need for further review; simply publish this manuscript. I am marking it as "accept with minor revisions" with the intent that the word "extremely" be inserted before "minor." Great job in a clear manuscript!
1. There appears to be irregular spacing before the in-text citations. Sometimes there is a space before the reference in brackets, sometime not. Please standardize.
2. The authors sometimes use the Oxford comma, sometimes not. Please standardize.
3. Line 47, change “..a assumed…” to “an assumed…”
4. The sentence that ends on line 51 should probably have a reference.
5. Line 63, change “…specificity ever…” to “…specificity even…”
6. Line 74, change “…by a MPX-specific…” to “…by an MPX-specific…”
7. Line 96, change “…three difference…” to “…three different…”
8. It seems the figure should have a citation because it came from a previous publication. It may even need copyright clearance from the original if that’s the case.
9. Line 148, I think the sentence would benefit from and addition like “…small breaks in the skin as can be seen in chicken pox lesions.”
10. Line 159, change “…tropical counties…” to “…tropical countries…”
11. Line 194, change “…subject was…” to “…subject were…”
12. Line 210, change “…for examples…” to “…for example…”
13. Line 216, change “…in the tonsil…” to “…in the tonsils…”
14. The last paragraph on page 5 would benefit from
a. Adding when pox lesions appear (line 215)
b. Outright stating whether IL-6 increased or decreased MPX pathogenesis, rather than simply affected it (lines 212 and 220)
15. Line 230, change “…why children…” to “…of why children”
Reviewer 2 Report
Khallafallah and Grose provide a review of the literature on evidence of coinfection of chickenpox and MPox; specifically in African children. The review relies heavily on 2 papers; [1] Hoff N et al. (2017) EcoHealth, [2] Hughes CM et al. (2021) Am. J. Trop. Med. Hyg. These two papers investigate the incidence of co-infection of VZV and MPX in patients.
Both Hoff et al. and Hughes CM et al. demonstrate there is a higher incidence of co-infection in patients between the age of 0-30. Hoff et al. report 13% of MPX infections (n= 151 of 1158) and Hughes et al. report 12.1% (n = 134 of 1107) are also co-infections with VZV. Both focusing on the Democratic Republic of Congo.
Major criticisms:
Reactivation of VZV due initial infection with MPX or other infections, stress, malnutrition etc. are poorly addressed, if not just brushed aside in the discussion (p6 Line 231-234). Reactivation of herpes viruses in children/adolescents/young adults is not unheard of in response to immunological and environmental stresses. This should be addressed further.
Vaccination rates of Hepatitis B, diptheria, tetanus, polio, VZV, measles, yellow fever, and rotavirus in the DRC have been declining in recent years and may account for enhanced susceptibility to infection and should be discussed.
Immune cross-protection between the vaccina virus immunisation (smallpox) and MPX should be discussed as a preventative measure.
Minor point:
WHO has recommended monkeypox be redefined to Mpox
Round 2
Reviewer 2 Report
I thank the authors for addressing my comments. It is an interesting take given the paucity of data coming out of the DRC and may provoke much needed interest in public health.
I would point out smallpox vaccine does show cross-protection against MPox (JYNNEOS) and could be included in the discussion as a preventative measure.